# Shifting the Focus: A Pilot Study on the Effects of Positive Body Exposure on Body Satisfaction, Body Attitude, Eating Pathology and Depressive Symptoms in Female Patients with Eating Disorders

**DOI:** 10.3390/ijerph191811794

**Published:** 2022-09-19

**Authors:** Marlies E. Rekkers, Lisanne Aardenburg, Mia Scheffers, Annemarie A. van Elburg, Jooske T. van Busschbach

**Affiliations:** 1Faculty of Social Sciences, Utrecht University, Heidelberglaan 1, 3584 CS Utrecht, The Netherlands; 2Department of Human Movement and Education, Windesheim University of Applied Sciences, Campus 2-6, 8017 CA Zwolle, The Netherlands; 3GGZ inGeest, Mental Health Institute, Laan van de Helende Meesters 433, 1186 DL Amstelveen, The Netherlands; 4Rintveld, Centre for Eating Disorders, Altrecht Mental Health Institute, 3705 WE Zeist, The Netherlands; 5University Center of Psychiatry, University Medical Center Groningen, University of Groningen, Rob Giel Onderzoekcentrum, Hanzeplein 1, 9713 GZ Groningen, The Netherlands

**Keywords:** eating disorders, attitudinal body image, body satisfaction, body attitude, body exposure, mirror exposure, depressive symptoms

## Abstract

One of the most commonly used techniques for the treatment of body image problems in eating disorders (ED) is body exposure (BE). However, evidence of its effectiveness in clinical populations is scarce. In the Positive Body Experience (PBE) protocol, the focus of positive BE is on aesthetic, functional and tactile aspects of the body. The current study evaluates the outcomes of positive BE with regard to changes in attitudinal body image and eating pathology, as well as the factors that influence these changes, in a sample of 84 adult female patients with different EDs who did not receive any other treatment for their EDs during the period in which BE treatment occurred. The results show significant positive changes in attitudinal body image, ED behaviors and depressive symptoms, with depressive symptoms at baseline mediating the changes in attitudinal body image. This study indicates that the PBE protocol is a suitable intervention for reducing negative attitudinal body image in anorexia and bulimia nervosa patients, as well as those with binge eating disorder. Furthermore, the results suggest that positive non-weight-related and functional body satisfaction are strong catalysts for change and that depressive symptoms play an important role in the ability to change. Additional RCTs are needed to gain more insight into the effects of PBE.

## 1. Introduction

Body image problems are a core feature of eating disorders (EDs) [1]. According to several authors, the different types of EDs, namely anorexia nervosa (AN), bulimia nervosa (BN), binge eating disorder (BED) and other specified eating and feeding disorder (OSFED), have the same maintaining mechanism: the over-evaluation of shape and weight and the control thereof, leading to body image problems. Such problems are a serious risk factor for the development and maintenance of EDs [2,3]. Therefore, targeting body image problems in treatment is crucial and associated with better overall treatment outcomes [4]. Moreover, the risk of relapse is high if body image problems are not adequately treated [5]. Therefore, research on interventions aimed at positively addressing body image problems has substantial public health significance [6].

Body image problems can be divided into perceptual and attitudinal body image problems [7,8]. The perceptual dimension refers to a disturbance in the perception of one’s own body [9]. The attitudinal dimension refers to affective, behavioral and cognitive components in the relationship with one’s own body. It manifests in body dissatisfaction and dysfunctional behavior, such as body-checking [10] and body avoidance [11], as well as on a cognitive level, e.g., criticizing and objectifying one’s body [12] and comparing one’s appearance negatively with that of others [13,14].

In the treatment of attitudinal body image problems, body exposure (BE) is a widely used approach [15,16,17]. During BE, patients stand in front of a mirror and look at their bodies while they are encouraged by the therapist to describe what they think and feel about their bodies [18]. BE has been shown to benefit individuals with high levels of body dissatisfaction and patients with EDs [16]. Tanck, Hartmann, Svaldi and Vocks [19] concluded that BE was effective at improving the affective, behavioral and cognitive components of a negative attitudinal body image.

Different variants of BE have been described [16]; they use three different cognitive restructuring approaches. In neutral BE, the subject is instructed to describe their reflection in the mirror, using non-judgmental descriptions of their appearance [20,21]. In pure BE, the emphasis lies on describing the thoughts and emotions that arise while looking at the negatively experienced body parts [22,23]. In positive BE, patients are encouraged to use language with positive valence while looking at their self-defined most attractive body parts [17,24,25]. Griffen, Naumann and Hildebrandt [16] reported on a small number of randomized trials in which these different approaches were compared using data from non-clinical groups composed of body-dissatisfied women [15,23,26,27]. In the comparison of non-judgmental BE and pure BE, both therapeutic techniques led to equal improvements with respect to positive and negative thoughts, but the pure variant was superior for reducing distress both within and between sessions [15,23]. Luethcke, McDaniel and Becker [27], found that positive BE was superior to non-judgmental descriptions in terms of reducing body dissatisfaction. Furthermore, Jansen et al. [26], and later Tanck et al. [19], concluded that positive BE seemed to be the more favorable option, as the pure variant resulted in heightened negative affect during BE and might therefore be experienced as more aversive. Furthermore, Griffen et al. [16] emphasized that the few clinical trials of BE have been small, with experimental designs, and that there is a great need for further clinical trials in specific ED groups (AN, BN and BED). Recently, Tanck et al. [25] also concluded that evidence on the effectiveness of different forms of BE in clinical populations is lacking.

Therefore, further research is needed, especially within a clinical context. This is particularly important since the severity of symptoms and co-morbidities may influence treatment results. In particular, depressive symptoms often co-occur with EDs [28,29] and have been found to be a predictor of body dissatisfaction in women with bulimia nervosa [30,31]. Furthermore, body dissatisfaction is associated with depressive symptoms [32], and Murray, Rieger and Byrne [33] found that people with depressive symptoms judge their bodies more negatively.

In this paper, a study is presented with the focus on positive BE using the ‘Positive Body Experience’ (PBE) protocol [17,34]. In this protocol, based on experimental research [35,36] and further non-clinical research into positive BE [18,24,26], guided self-confrontation with the help of a mirror is the key element. During this self-confrontation, patients are instructed to describe their positively experienced body parts in a positive way and to refrain from looking at or speaking about their negatively experienced body parts. In addition, the PBE protocol not only addresses the aesthetic aspects of positively experienced body parts but also pays attention to positive functional and tactile aspects. These aspects of positively experienced body parts could serve as an important ingredient of positive body image [1,17].

The approach used in the PBE protocol was based on studies that show that a positive functional perception of the body or body parts can serve as a protective psychological mechanism against body dissatisfaction [37,38,39,40,41]. Patients with EDs tend to focus on aesthetic aspects when they evaluate their bodies and base body satisfaction on physical attractiveness. To help improve body image, it is important to broaden this perspective and shift attention from the way the body looks to the functional aspects of the body, i.e., what the body can do [42].

The objective of the current pilot study was to evaluate the results of the PBE protocol in female patients with EDs. A three-way approach was chosen: The first aim was to assess the change in body image after participating in PBE and whether this change was clinically relevant. The second aim was to explore whether patients with different EDs benefitted differently from the PBE protocol. We hypothesized that patients with AN and BN would profit most because body image problems are explicitly mentioned in the DSM classifications for these conditions, which is not the case for BED. The third and last aim was to examine the factors that influenced changes in attitudinal body image. We assumed that the severity of the ED before the start of treatment would negatively mediate the extent of the change in body image found post-treatment. The second assumed factor was the severity of depressive symptoms before treatment. Since ED recovery is associated with the absence of major depressive symptoms [43], we assumed that the presence of these symptoms might also negatively mediate the treatment of body image problems.

## 2. Materials and Methods

### 2.1. Participants, Design and Procedure

Participants were female patients attending an outpatient clinic specialized in the treatment of EDs in the Netherlands between January 2010 and June 2021. All participants were treated for their negative body image with the PBE protocol and had a primary diagnosis of an ED (AN, BN, BED or OSFED), which was assessed by an experienced clinician based on DSM-IV criteria before 2017 and DSM-5 criteria from 2017 onwards. From 2010 to 2017, BED was separately diagnosed within the Eating Disorder Not Otherwise Specified (EDNOS) group. During the treatment for negative body image, no other treatment targeting ED pathology or body image took place. In general, sessions took place once every two weeks. As this study was designed as a pilot test of the primary changes after treatment with the PBE protocol, a single-arm pretest/posttest design was used.

Only Dutch-speaking participants over 18 were included. From a sample of 121 patients, the data of 13 patients were removed for the following reasons: younger than 18 (*n* = 3); English-speaking (*n* = 5); no permission for research (*n* = 5). From the remaining sample (*n* = 108), 24 patients had no post-treatment data for a variety of reasons: premature termination of sessions because of changes in residence, work or study (*n* = 14); referred to a more intensive treatment for their ED (*n* = 3) or for comorbid disorders (*n* = 3); dissatisfaction with the treatment (*n* = 2); financial circumstances (*n* = 1); reasons unknown (*n* = 2). This resulted in a clinical sample of 84 female participants with pre- and post-measurements. In this sample, 26 women (31%) had a diagnosis of AN, 16 (19%) BN, 11 (13%) BED and 31 (37%) OSFED.

According to the Dutch law on medical scientific research with human subjects, all patients participating in this study signed an informed consent form with a standard format, as prescribed by the Dutch Central Committee on Research Involving Human Subjects (CCMO) (see www.ccmo.nl, accessed on 1 September 2022). The questionnaires the participants filled out and the treatment the participants followed were both components of treatment as usual (TAU). This means there were no invasive interventions. In such cases, ethical approval of the CCMO or another specialized external committee is not required in the Netherlands.

The assessment battery consisted of four questionnaires, two for measuring attitudinal body image (*n* = 84), one for measuring ED pathology (*n* = 63) and one for measuring depression (*n* = 78). There were no missing data in the completed questionnaires. The questionnaires were filled out on the participants’ own time on private devices. Six participants preferred to fill out the questionnaires on paper.

### 2.2. Measures

The Body Cathexis Scale (BCS) [44]; Dutch version: Dorhout, Basten, Bosscher and Scheffers [45] measures the degree of satisfaction with the appearance and functionality of different parts of the body. The BCS consists of 40 items rated on a 5-point Likert scale (from 1 = ‘very dissatisfied’ to 5 = ‘very satisfied’). Higher scores indicate a higher level of body satisfaction. The construct and concurrent validity of the original scale is good [46,47,48]. Research on the Dutch version of the BCS in both clinical (*n* = 238) and non-clinical (*n* = 1060) samples revealed three subscales: functional body satisfaction, weight-related body satisfaction and non-weight-related body satisfaction [49]. Internal consistency was adequate for both samples, with Cronbach’s α = 0.90 for the total scale and Cronbach’s α = 0.83–0.85 for the subscales in the clinical sample [49].

The Body Attitude Test (BAT) [50] measures subjective body experience and attitude towards one’s own body. The BAT consists of 20 items rated on a 6-point Likert scale (range 0–5). The maximum score is 100, and the higher the score, the more body attitude deviates from that of the general population. The internal consistency is good, with Cronbach’s α = 0.93; test–retest reliability is also good (ICC = 0.92), as are convergent and divergence validity [50,51,52,53].

The Eating Disorder Examination Questionnaire (EDE-Q) [54]; Dutch version: Aardoom, Dingemans, Slof Op’t landt and Van Furth [55] measures ED symptoms. The EDE-Q consists of 36 items, of which 22 determine the total score. These 22 items comprise four subscales, assessing restraint, shape concern, weight concern and eating concern over the previous 28 days; questions are answered on a 7-point Likert scale ranging from 0, ‘not one day’, to 6, ‘every day’. Higher scores are indicative of higher levels of ED psychopathology. The construct validity and internal consistency (with a Cronbach’s alpha of 0.95 for the total scale and values of Cronbach’s alpha varying from 0.81 to 0.91 for the subscales) of the Dutch version are good [55].

The Beck Depression Inventory-II (BDI-II) [56]; Dutch version Van der Does [57] measures the degree of depressive symptoms. The BDI-II consists of 21 items rated on a 4-point Likert scale (ranging from 0 = symptoms absent to 3 = severe symptoms). Total scores range between 0 and 63, with higher scores reflecting higher levels of depression. The reliability and validity of the Dutch version of the BDI-II have been supported by the findings of previous works [57,58].

The recorded demographics were age, height, weight and level of education at pre- and post-treatment. In addition, the number of sessions that took place was monitored.

### 2.3. The Positive Body Experience Protocol (PBE)

The PBE protocol (for a more thorough description and an overview of the theoretical foundation, see Rekkers et al. [17] (in English) and Rekkers and Van Gullik [34] (in Dutch)) includes four phases and can be used in both individual and group treatment. In the first and the second phases, *the goals* are to increase knowledge about one’s dysfunctional body experience (phase one) and to receive psycho-education about healthy body image (phase two). These issues are important because they are a prerequisite for successful body exposure. The actual BE takes place in the action phase (phase three), with a maximum of eight sessions. Both self-confrontation with the help of a mirror and hetero-confrontation using comparison exercises are key elements of BE. Hetero-confrontation refers to looking at images of others and, in the case of BE, comparing oneself positively with these images (downward comparison). In the fourth and last phase, the treatment focus lies on stabilization, including the repetition of BE and relapse prevention. The total number of sessions in the PBE protocol can vary between 12 and 17 sessions, depending on how much psycho-education before BE and how much stabilization after the action phase are required.

### 2.4. Statistical Analysis

Descriptive analyses of the characteristics of the participants, the length of the intervention and its outcomes are presented in percentages, means and standard deviations, respectively. Associations between body satisfaction, body attitude, ED pathology and depressive symptoms at the start were analyzed using Pearson correlations to enhance insight into the data; correlations were considered large if the value of *r* was between 0.50 and 1.0, medium if it was between 0.29 and 0.49 and small if it was between 0.10 and 0.29 [59].

In order to assess the outcome of the PBE protocol, repeated measures analyses of variance were calculated with ‘time’ (pre- and post-intervention) as the within-subject factor. This was performed four times using data from the total group, with body satisfaction (BCS), body attitude (BAT), ED pathology (EDE-Q) and depression scores (BDI) as the dependent variables. Additionally, the clinical relevance of the changes found was determined using the MCID [60,61], defined as the standard deviation of the theoretical total distribution from the primary outcome multiplied by 0.5. Cohen’s *d* was used to determine effect sizes in the univariate tests and was considered large if *d* > 0.80, medium if the value of *d* was between 0.50 and 0.79 and small if its value was between 0.20 and 0.49 [62].

As a second step, analyses were conducted that focused specifically on the subgroups of participants with different types of ED diagnoses. To obtain a better picture of the differences in treatment effects for the various diagnostic ED groups, the 31 participants with a primary diagnosis of OSFED were, according to their clinical presentations, added to one of the three other diagnostic groups: AN (previous *n* = 26; new total group *n* = 46, 55%), BN (previous *n* = 16; new total group *n* = 25, 30%) and BED (previous *n* = 2; new total group *n* = 11, 15%). Analyses were performed focusing on the differences between the three groups in terms of both characteristics and pre- and post-intervention measurements. To compare the extent of the changes following positive body exposure between the different diagnostic groups, change scores (post-measure minus pre-measure) were calculated for BCS, BAT and EDE-Q; in addition, an ANOVA test of variance was conducted. In both sets of analyses, Levene’s test was used to test for the homogeneity of the variances for all three groups. Depending on the outcome, the Tukey post hoc test or the Games–Howell post hoc test was chosen to gain insight into which of the groups could account for the differences between the groups. Sensitivity analyses were conducted to compare the outcomes with the outcomes of the ANOVA analyses using the original four diagnostic groups.

The third step in the analysis encompassed a mediation analysis, with depressive symptoms (BDI-II) and the severity of the ED (EDE-Q) at the start of therapy as mediators of the relationship between the pre- and post-measurement outcomes; body satisfaction (BCS) and body attitude (BAT) were analyzed separately. To perform the mediation analyses, the PROCESS Macro in SPSS, which was developed by [63] Preacher and Hayes (2004), was used.

In all analyses, a value of *p* < 0.05 was considered significant. Data were controlled for input errors and normality. All data were analyzed using IBM SPSS version 24.0.

## 3. Results

### 3.1. Characteristics of the Sample, Number of Sessions and Primary Outcomes

The mean age of the sample (84 female participants) was 26.67 (*SD* = 5.09, range = 19–46). Most participants had a moderate (28.6%) or high (63.1%) level of education; 8.3% of the participants had a lower level of education. Table 1 shows the characteristics of the total group and the three diagnostic groups, namely AN, BN and BED. The mean age of the BED group was higher than those of the AN and BN groups (Tukey post hoc test *p* = 0.035 and 0.037, respectively). The mean BMI was 22.55 (*SD* = 4.2) at baseline and 22.60 (*SD* = 4.2) at discharge. As could be expected, there were significant differences between participants in the three diagnostic groups for both BMI measures, with values for the AN group significantly lower than those for the BN group (*p* = 0.02 for pre-treatment BMI and *p* = < 0.01 for post-treatment BMI); the BN group, in turn, had values significantly lower than those of the BED group (*p* < 0.01 for both measures). The mean BMI values for the original four diagnostic groups can be found in the note of Table 1. Table 1 also shows the mean number of sessions of the PBE protocol that participants received. A total of 25% of the participants received 11 sessions or less, 25% 11 to 13 sessions, 25% 13 to 17 sessions and 25% more than 17, with a maximum of 27. The participants with AN received the highest average number of sessions. This average was significantly more than the average for the participants with BN (*p* = 0.03), who received the lowest average number of sessions. There was no significant difference in the average number of sessions between participants with AN and BED or between those with BN and BED.

Table 2 shows the results of the primary outcomes and possible mediators for the total sample and the three diagnostic groups. With regard to attitudinal body image, participants with AN showed higher BCS-Weight scores at baseline than participants with BED (*p* = 0.03), with no significant differences between the AN and BN groups (*p* = 0.07) or between the BN and BED groups (*p* = 0.68). For the AN group, BAT scores were also significantly lower than for the BED group (*p* < 0.01); again, there were no significant differences between the AN and BN groups (*p* = 0.07) or between the BN and BED groups (*p* = 0.73). With regard to the baseline scores for the EDE-Q total, the EDE-Q subscale for eating concern and the EDE-Q subscale for restraint, participants with BN had significantly higher scores than participants with AN (*p* = 0.04, *p* < 0.01 and *p* = 0.02, respectively). The differences between the BN and BED groups were only significant for the EDE-Q subscale for restraint (*p* < 0.01). No significant differences in regard to the EDE-Q subscale for weight concern or the EDE-Q subscale for shape concern were found between the three diagnostic groups.

### 3.2. Correlations between Pre-Treatment Measures

At pre-treatment, almost all outcomes were significantly correlated, with correlations varying from small to large (Table 3). There were five exceptions. BCS total score, BCS-Non-weight and BCS-Functional had no or small non-significant associations with ED symptoms (EDE-Q); BCS-Functional had no or small non-significant associations with BCS-Weight and Body attitude (BAT).

### 3.3. Differences in Outcomes for the Whole Group

In order to assess the outcomes of the PBE protocol, repeated measures analyses of variance were calculated with ‘time’ (pre- and post-intervention) as the within-subject factor. There was a statistically significant difference over time for all outcomes, namely body satisfaction (BCS), including all subscales; body attitude (BAT); ED pathology (EDE-Q), including all subscales; and depressive symptoms (BDI). The clinical relevance of all the observed changes was significant according to the MCID (see Table 4).

### 3.4. Differences in Outcomes for the Three Diagnostic Groups

To compare the changes over time between the three diagnostic groups for all outcome measures, a one-way ANOVA was performed, with the change scores (differences post-intervention minus pre-intervention) calculated for the BCS, BAT, EDE-Q and BDI scales (see Table 5). There was only one statistically significant difference in the pre–post change between the three diagnostic groups, which was for the outcome measure BAT *F* (2.81) = 3.715, *p* = 0.029. A Games–Howell post hoc test revealed that the extent of change was significantly lower for the AN group (*M* = −15.37, *p* = 0.04) than for the BED group (*M* = −25.00). There were no statistically significant differences between the BN group and the BED group (*p* = 0.781) or the AN group (*p* = 0.128). For the sensitivity analyses, one-way ANOVAs were performed on the change scores a second time, but instead of three groups, the original four diagnostic groups, including one OSFED group, were used. These analyses showed comparable results, with no significant differences between the four diagnostic groups for the BCS, EDE-Q or BDI scales. The results were different with regard to changes in BAT scores: in the analyses using three groups, lower scores were found for the AN group, but in the analyses using four groups, no between-group differences were found (*F* = 1.29, *p* = 0.285).

### 3.5. Mediation Analyses

The results of the mediation analysis with change in BCS as the dependent variable showed that the relationship between the initial and final measurements on the BCS increased from β = 0.421 to β = 0.576 after adding depressive symptoms as a mediator. Depressive symptoms at baseline were a significant mediator of the relationship between body satisfaction at baseline and body satisfaction at the end of therapy: β = 0.621, *t* = 5.612, *p* < 0.01. This model was found to be significant: *F* (2.76) = 18.909, *p* < 0.001. The explained variance of the model was 17.9% before mediation and 33.2% after adding the mediation. The relationship between the initial and final measurements on the BAT increased from β = 0.353 to β = 0.602 after adding depressive symptoms as a mediator. Depressive symptoms at baseline were also a significant mediator in the relationship between body attitude at baseline and body attitude at the end of therapy: β = 0.061, *t* = 5.322, *p* < 0.01. This model was found to be significant: *F* (2.76) = 21.64, *p* < 0.001. There was partial mediation since the relationships between the initial and final measurements on the BCS and BAT without a mediator were also significant.

Results showed that the severity of the ED at baseline was not a significant mediator in the relationship between body satisfaction at baseline and at the end of therapy: β = 0.045, *t* = 0.063, *p* = 0.479. In addition, the severity of the ED at baseline was not a significant mediator in the relationship between body attitude at baseline and at the end of therapy: β = −1.125, *t* = −0.681, *p* = 0.499.

## 4. Discussion

This explorative study evaluated the treatment of negative body image using the PBE protocol in female participants with EDs in a clinical setting. Results show that positive body exposure led to significant positive changes in attitudinal body image, with large effect scores. In addition, eating pathology and depressive symptoms show a significant decrease, with large effect scores. For clinical practice, it is important to look into the clinical relevance of research results, because statistical significance does not necessarily mean clinical relevance [61]. According to the MCID, all difference scores for all outcome measures were clinically relevant, because all average difference scores exceeded the MCID threshold. This implies that, for the average participant, there was a clinically relevant difference in the improvement of attitudinal body image and the reduction in eating pathology and depressive symptoms after following the PBE protocol.

A remarkable finding was that the BCS-Weight subscale scores were also significantly higher after following the PBE protocol. During the body exposure treatment, the participants were instructed to focus exclusively on positively experienced body parts and to refrain from looking at or speaking about negatively experienced body parts, which are often weight-related body parts. A possible explanation for the increase in satisfaction for weight-related body parts may relate to the assumption that when positively experienced body parts are more prominent, the focus on negatively experienced weight-related body parts may shift to the background, which could result in weight-related body dissatisfaction being experienced as less negative by the patient. In addition, there were no correlations found in this study between eating pathology and non-weight-related or functional body satisfaction. This lack of coherence may be clinically relevant as a possible motivation for initiating change. Looking at the present study, mitigating weight-related body dissatisfaction by focusing on non-weight-related and functional body satisfaction [64,65] seemed to work.

Another noteworthy finding was that the score changes for all EDE-Q subscales, including the subscales for restraint and eating concern, which are related to eating behavior, were significant. These results imply that the treatment of attitudinal body image with positive BE alone can create a significant reduction in disturbed eating behavior. It also raises an interesting question as to whether body image should be regarded as one of the sources of EDs, and not only a maintaining factor thereof, following the definition of Fairburn et al. (2003). In this context, Phillipou, Castel and Rossell [66] also posed the question of whether the conceptualization of AN and BN as ‘EDs’ was simplistic and misleading; they even argued for another classification system, classifying AN and BN not as EDs but as body image disorders. This would, according to the authors, result in more emphasis being placed on body image problems, leading to fewer misperceptions and comments focused on eating behavior, along with a change in the research agenda concerning EDs. Regardless of whether or not a new classification is considered, the results of the EDE-Q in this study support the notion that the effective treatment of body image problems is probably essential in the treatment of EDs [67,68].

Contrary to expectations, there was almost no difference in the effectiveness of positive BE for patients with different EDs. The only exception was the change in body attitude, as measured by the BAT, which was significantly larger for the BED group than for the AN group. Even the fact that the BED group benefitted from positive body exposure is noteworthy, because body image problems are not explicitly mentioned as a diagnostic criterion in the DSM-5 classification of BED, in contrast with AN and BN. Nevertheless, Lewer, Bauer, Hartmann and Vocks [69] found in their narrative review that attitudinal body image problems, such as body dissatisfaction, overconcern with weight and shape, body-related checking and avoidance behavior, also occur in BED. In our sample, we see that the body dissatisfaction scores of the participants with BED corresponded with those of AN and BN participants; for body attitude, as measured with the BAT, BED scores were even more negative compared with those of the AN and BN groups. In line with these findings, Krohmer, Naumann, Tuschen-Caffier and Svaldi [70] stated that there is growing evidence that body image problems also play an important role in the development and maintenance of BED. At the same time, Lewer et al. [69] concluded that research on treatments focusing directly on body image problems in BED is still scarce. This study contributes to knowledge on how to improve therapeutic options for BED.

We examined two factors that possibly influenced the change in attitudinal body image. We assumed that the severity of these two factors at pre-treatment would reduce the change in body image post-treatment. The first factor was the severity of the ED. It was surprising that this factor had no influence. A possible explanation is that this result is related to the aforementioned assumption that EDs may be the result of a body image disorder [66]. In addition, Alleva, Martijn, Van Breukelen, Jansen and Karos [71] stated that the severity of body image problems is associated with the persistence of EDs.

The second factor was the severity of depressive symptoms. Two separate mediation analyses showed that depressive symptoms meditated the relationship between pre- and post-intervention scores for body attitude (BAT) but not for body satisfaction (BCS). The more severe the depressive symptoms, the smaller the change in body attitude. Body attitude refers to the cognitive, affective and behavioral attitudes towards the body [72], and body satisfaction refers to the degree of contentment with the appearance or functionality of the body [46]. In line with our results, Van Mierlo, Scheffers and Koning [73] found that body attitude appears to be a somewhat stronger predictor of depressive symptoms than body satisfaction. Other literature also shows that there is a reciprocal relationship between depressive symptoms and negative attitudinal body image. On the one hand, a negative attitudinal body image can be a risk factor for the development of a mood disorder [74], while on the other hand, the body is judged less positively if there is a mood disorder [33].

In this study, depressive symptoms only mediated the results for the BAT; a possible explanation could be the hypothesis that, as a construct, body attitude is more comprehensive than body satisfaction. Before, during and after BE in the PBE protocol, the treatment is aimed not only at encouraging affective appreciation of the body (body satisfaction) but also at challenging dysfunctional thoughts and behaviors related to body image [17]. It could be that a high severity of depressive symptoms at the start of the treatment makes it more difficult to challenge and change these dysfunctional thoughts and behaviors. Griffen et al. [16] reported in their review that, in several studies, dropouts from BE treatment had significantly higher baseline depression scores than other participants; the authors concluded that caution was warranted when treating individuals with a history of self-injurious behavior or current clinical depression. Our findings confirmed this; we showed that higher severity depression symptoms negatively influenced the outcome of the intervention. We, therefore, advocate for special attention to depressive symptoms when patients are referred to BE; if needed, extra forms of treatment should be offered.

This clinical trial must be interpreted in light of several limitations that should inform future positive BE research. The lack of a control group and follow-up measures implies that the results must be interpreted with caution. Although a strength of this study is the fact that participants did not receive any other treatment for their ED during the period in which positive BE was administered, it is still unclear whether positive BE is a better option, compared to other variants of BE, in a clinical setting. Future studies should therefore implement clinical randomized controlled designs in order to compare different variants of BE or to evaluate positive BE versus other interventions that aim to improve aspects of body image and eating symptomatology. Concerning follow-up measures, Khalsa, Portnoff, McCurdy-McKinnon and Feusner [75] emphasized the importance of these measures in order to analyze the stability of effects as well as relapses in the context of EDs. However, in line with the experiences of Tanck et al., [25] in their clinical research on positive BE, we could not collect follow-up data due to the fact that participants in an outpatient setting are discharged after treatment. Future research should therefore consider how to implement follow–up measurements, including in an outpatient treatment setting, in order to analyze effect stability.

Another limitation is the generalizability of this study. Since the positive BE took place in an outpatient setting, the potential severity of the eating pathology may differ from what is typical for inpatient treatment. The average (*M* = 3.51, *SD* = 0.89) of the EDE-Q results in the present study for the total sample at pre-test were lower compared to a Dutch clinical inpatient sample (*n* = 935, *M* = 4.02, *SD* = 1.28) [55]. In addition, the mean BMI at pre-test for the AN group in the present study was relatively high (*M* = 20.53, *SD* = 2.57) and probably differed from a clinical inpatient setting. A recommendation for future research is to investigate the positive BE protocol in a group consisting of inpatient participants, while taking into account whether the severity of underweight observed for AN participants influences the effects of positive BE.

All participants received positive BE according to a fixed protocol, i.e., the PBE protocol [17,34]. An important recommendation by Jansen et al. (2013) [18] is that offering BE according to a structured and detailed protocol is more effective than having patients look at their own bodies in the mirror without a clear purpose. For this reason, it is crucial that BE protocols are available for clinical practice and further research. It may be considered a shortcoming that not all participants in this study received the same number of sessions. However, because no clinical trial has empirically determined the ideal length of BE [16], we decided, as a first step, to explore the question of whether the number of sessions participants received corresponded with the prescribed number of sessions (12–17) in the PBE protocol. The results indicate that the average number of sessions was 14.15 for all participants, of whom 75% received a maximum of 17 sessions.

Finally, it is important to mention that all participants were female, predominantly highly educated and, on average, in their twenties. Although these characteristics are representative for EDs, and the overrepresentation of patients with high levels of education is often observed [76,77], it is unclear whether our findings can be transferred to men and midlife women with an ED, in cases where body image problems are also present. Due to a lower prevalence of EDs, these groups receive little attention [78,79]. Future research should, therefore, investigate the effects of positive BE in men and older women with an ED.

## 5. Conclusions

The present study made it clear that positive BE, administered with the PBE protocol, leads to significant positive changes in attitudinal body image, eating pathology and depressive symptoms in female participants with EDs (AN, BN and BED) in a clinical setting. Moreover, all difference scores on all outcome measures were clinically relevant. These results indicate that positive BE is a suitable intervention for reducing negative attitudinal body image not only for patients with AN and BN but also for those with BED. Furthermore, results indicate that mitigating weight-related body dissatisfaction by learning to shift the focus to positive non-weight-related and functional body satisfaction is a strong catalyst for change. In contrast, depressive symptoms were found to be a negative mediator: more severe depressive symptoms reduced the change in body attitude. While the results must be interpreted with caution because of a lack of a control group and follow-up measures, this study certainly expands the available knowledge about the effectiveness and effect of positive BE in a clinical setting.

## Figures and Tables

**Table 1 ijerph-19-11794-t001:** Characteristics of the total sample and the three diagnostic groups.

	Total (*n* = 84) *M (SD)*	AN (*n* = 46) *M (SD)*	BN (*n* = 25) *M (SD)*	BED (*n* = 13) *M (SD)*	*F*
Age	25.97 (5.02)	25.49 (4.35)	25.01 (4.46)	29.51 (5.02)	3.72 *
BMI—pre #	22.55 (4.21)	20.53 (2.57)	22.38 (2.91)	30.01 (4.21)	66.58 **
BMI—post	22.64 (4.20)	20.58 (2.44)	22.42 (2.60)	30.25 (2.60)	74.97 **
Sessions	14.15 (4.71)	15.26 (4.86)	12.32 (4.45)	13.77 (3.68)	3.40 *

*Note: M* = mean, *SD* = standard deviation, AN = anorexia nervosa, BN = bulimia nervosa, BED = binge eating disorder, BMI = body mass index, * *p* < 0.05, ** *p* < 0.01. # Mean BMI values for the original four diagnostic groups: AN (*M* = 19.03, *SD* = 1.18), BN (*M* = 22.64, *SD* = 3.44), BED (*M* = 30.57, *SD* = 1.76) and OSFED (*M* = 22.59, *SD* = 2.55).

**Table 2 ijerph-19-11794-t002:** Pre- and post-treatment outcomes and possible mediators for the total sample and the three diagnostic groups, with ANOVA results for the between-group differences.

	Pre-Treatment					Post-Treatment				
	Total (*n* = 84)	AN (*n* = 46)	BN (*n* = 25)	BED (*n* = 13)	*F*	Total (*n* = 84)	AN (*n* = 46)	BN (*n* = 25)	BED (*n* = 13)	*F*
BCS total	2.87 (0.46)	2.91 (0.47)	2.85 (0.45)	2.83 (4.99)	0.19	3.37 (0.50)	3.34 (0.53)	3.44 (0.43)	3.39 (0.54)	0.34
- BCS-NW	3.16 (0.55)	3.17 (0.59)	3.13 (0.47)	3.13 (0.57)	0.05	3.62 (0.48)	3.57 (0.49)	3.72 (0.46)	3.39 (0.51)	0.88
- BCS-W	1.99 (0.69)	2.18 (0.70)	1.82 (0.64)	1.63 (0.61)	4.74 *	2.67 (0.78)	2.77 (0.80)	2.64 (0.74)	2.49 (0.81)	0.73
- BCS-F	2.92 (0.61)	2.86 (0.49)	2.98 (0.71)	3.50 (0.73)	0.57	3.41 (0.61)	3.32 (0.59)	3.54 (0.57)	3.49 (0.73)	1.24
BAT	61.66 (15.05)	57.21 (15.13)	65.16 (12.99)	72.61 (11.80)	7.01 **	43.07 (14.38)	41.84 (15.79)	42.98 (10.91)	47.62 (15.36)	0.82
EDE-Q total #	3.51 (0.88)	3.29 (0.87)	3.90 (0.92)	3.66 (0.64)	3.17 *	2.31 (0.97)	2.25 (1.03)	2.40 (0.96)	2.38 (0.82)	0.16
- EDE-Q-R	2.74 (1.16)	2.58 (1.22)	3.39 (0.92)	2.28 (0.88)	4.20 *	1.61 (1.06)	1.55 (1.10)	1.68 (0.98)	1.68 (1.14)	0.12
- EDE-Q-EC	2.66 (1.28)	2.21 (1.03)	3.49 (1.51)	2.86 (0.99)	7.20 *	1.57 (0.97)	1.43 (1.05)	1.81 (0.86)	1.64 (0.80)	0.91
- EDE-Q-WC	4.19 (1.02)	4.02 (1.10)	4.25 (0.91)	4.72 (0.71)	1.97	2.90 (1.04)	2.82 (1.06)	2.94 (1.07)	3.10 (0.99)	0.31
- EDE-Q-SC	4.58 (1.02)	4.44 (1.02)	4.70 (1.11)	4.87 (0.88)	0.84	3.08 (1.30)	3.01 (1.31)	3.21 (1.42)	3.11 (1.67)	0.14
BDI-II #	17.02 (9.16)	16.76 (8.70)	17.46 (10.53)	17.09 (8.59)	0.04	9.33 (7.26)	10.50 (8.67)	7.44 (5.28)	9.04 (2.94)	1.43

*Note:* AN = anorexia nervosa, BN = bulimia nervosa, BED = binge eating disorder, BMI = body mass index, BCS = Body Cathexis Scale, BCS-NW = Body Cathexis Scale-Non-Weight, BCS-W = Body Cathexis Scale-Weight, BCS-F = Body Cathexis Scale-Functionality, BAT = Body Attitude Test, EDE-Q = Eating Disorder Examination Questionnaire, EDE-Q-R = Eating Disorder Examination Questionnaire-Restraint, EDE-Q-EC = Eating Disorder Examination Questionnaire-Eating Concern, EDE-WC = Eating Disorder Examination Questionnaire-Weight Concern, EDE-SC = Eating Disorder Examination Questionnaire-Shape Concern, BDI-II = Beck Depression Inventory-II, * *p* < 0.05, ** *p* < 0.01. # For EDE-Q, *n* = 63 (AN 36, BN 17 and BED 10), and for BDI-II, *n* = 78 (AN 44, BN 24 and BED 10).

**Table 3 ijerph-19-11794-t003:** Correlation matrix for the pre-treatment measurements.

Measures	BCS-NW	BCS-W	BCS-F	BAT	EDE-Q #	BDI-II #
BCS-total	0.853 **	0.639 **	0.606 **	−0.413 **	−0.236	−0.420 **
BCS-NW		0.421 **	0.470 **	−0.220 *	0.004	−0.244 *
BCS-W			0.043	−0.568 **	−0.416 **	−0.286 **
BCS-F				−0.143	0.045	−0.413 **
BAT					0.386 **	0.492 **
EDE-Q						0.343 **

*Note*: BCS = Body Cathexis Scale, BAT = Body Attitude Test, EDE-Q = Eating Disorder Examination Questionnaire, BDI-II = Beck Depression Inventory-II, BCS-NW = Body Cathexis Scale-Non-Weight, BCS-W = Body Cathexis Scale-Weight, BCS-F = Body Cathexis Scale-Functionality, * *p* < 0.05, ** *p* < 0.01. # For EDE-Q, *n* = 63; for BDI-II, *n* = 78.

**Table 4 ijerph-19-11794-t004:** Means (*M*) and standard deviations (*SD*) of the differences between pre- and post-intervention scores for all outcomes, results of repeated measure analyses (*F*) and minimal clinically important difference (*MCID*), test of the difference and effect sizes (Cohen’s *d*).

Measures	Mean Differences *(SD)*	*F*	*SEM*	*MCID*	*t*	Cohen’s *d*
BCS Total (*n* = 84)	0.50 (0.45)	101.34 **	0.05	0.21	−10.07 **	1.04
BCS-NW	0.46 (0.52)	34.85 **	0.06	0.26	−9.08 **	0.89
BCS-W	0.70 (0.71)	82.47 **	0.08	0.36	−8.24 **	0.92
BCS-F	0.49 (0.58)	61.01 **	0.06	0.29	−7.81 **	0.80
BAT Total (*n* = 84)	−18.89 (13.79)	157.60 **	1.50	0.78	12.55 **	1.28
EDE-Q Total (*n* = 63)	−1.20 (1.13)	71.49 **	0.14	0.57	8.45 **	1.29
EDE-Q-R	−1.14 (1.52)	35.68 **	0.19	0.76	5.79 **	1.52
EDE-Q-EC	−1.09 (1.29)	44.68 **	0.16	0.65	6.68 **	0.96
EDE-Q-WC	−1.30 (1.28)	64.53 **	0.16	0.64	8.03 **	1.25
EDE-Q-SC	−1.50 (1.34)	79.11 **	0.17	0.67	8.89 **	1.28
BDI-II Total (*n* = 78)	−7.76 (7.76)	77.58 **	0.87	3.88	8.81 **	0.93

*Note*: BCS = Body Cathexis Scale, BCS-NW = Body Cathexis Scale-Non-Weight, BCS-W = Body Cathexis Scale-Weight, BCS-F = Body Cathexis Scale-Functionality, BAT = Body Attitude Questionnaire, EDE-Q = Eating Disorder Examination Questionnaire, EDE-Q-R = Eating Disorder Examination Questionnaire-Restraint, EDE-Q-EC = Eating Disorder Examination Questionnaire-Eating Concern, EDE-WC = Eating Disorder Examination Questionnaire-Weight Concern, EDE-SC = Eating Disorder Examination Questionnaire-Shape Concern, BDI-II = Beck Depression Inventory-II, *SEM* = standard error of the mean, *t* = t-score, ** *p* < 0.01.

**Table 5 ijerph-19-11794-t005:** ANOVA with mean difference scores for the outcome measures BCS, BAT, BDI and EDE-Q for the different diagnostic groups.

	AN *M (SD)* (*n* = 46)	BN *M (SD)* (*n* = 25)	BED *M (SD)* (*n* = 13)	Total *M (SD)* (*n* = 84)	*F*
BCS	0.43 (0.47)	0.58 (0.44)	0.56 (0.41)	0.9 (0.45)	1.10
BAT	−15.37 (13.52)	−22.18 (14.02)	−25.00 (11.28)	−18.89 (13.79)	3.72 *
BDI #	−6.26 (7.45)	−10.02 (8.10)	−8.31 (7.60)	−7.69 (7.76)	1.91
EDE-Q #	−1.04 (1.11)	−1.50 (1.27)	−1.28 (0.88)	−1.20 (1.13)	1.01

*Note:* AN = anorexia nervosa, BN = bulimia nervosa, BED = binge eating disorder, BCS = Body Cathexis Scale, BAT = Body Attitude Questionnaire, EDE-Q = Eating Disorder Examination Questionnaire, BDI-II = Beck Depression Inventory-II. # For EDE-Q, *n* = 61 (AN 36, BN 17 and BED 10), and for BDI-II, *n* = 78 (AN 44, BN 24 and BED 10). * *p* < 0.05.

## Data Availability

Not applicable.

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
