# Peer review of "Shifting the Focus: A Pilot Study on the Effects of Positive Body Exposure on Body Satisfaction, Body Attitude, Eating Pathology and Depressive Symptoms in Female Patients with Eating Disorders"

_ijerph, 2022, doi:10.3390/ijerph191811794_

Round 1
Reviewer 1 Report
This manuscript describes an outpatient pilot trial of positive body exposure in women with eating disorders. This was not a randomized controlled trial. The intervention was shown to result in improvement in eating disorder psychopathology, depressive symptoms, and body image attitude. Baseline depression was determined to be a mediator of treatment outcome.
Body image distress is a hallmark feature of several of the eating disorders and the topic of effective interventions for this and related symptoms is an important one that has only fairly recently garnered increased attention in behavioral research. However, there are several aspects of the study’s methodology and manuscript in need of revision and/or additional clarification.
1. First, a few questions about the sample.
Though the authors describe their sample as a multi-diagnostic group, they do not highlight that the mean BMI of their AN group was around 20.5 kg/m2. This suggests that these participants were weight-restored and it would be useful to understand how recently these individuals had normalized their weight.
Why were men not included in the study?
2. With regard to the study intervention, it would be useful to elaborate on what is meant by “hetero-confrontation, using comparison exercises” (pg. 4, section 2.3) as this is a “key element” of the treatment.
The first sentence of section 2.3 is stated twice consecutively (typo).
The authors prudently point out in their discussion that a limitation of the study was that the mean number of sessions received was highly variable. The protocol is described as varying between 12-17 sessions, but a quarter of the sample received more than 17 sessions and a quarter of the sample received less than 12 sessions. Why was this? Did all participants finish all four phases of the protocol as outlined in section 2.3? Was the number of sessions received considered as a potential covariate in analyses? Was feasibility and acceptability data collected (and if so, please include in results)?
3. Table 2 indicates the mean(SD) of pre-treatment EDE-Q-R scores for the BN group to be 5.19(7.48). This SD seems likely to be incorrect since the scoring for EDE-Q items is 0-6. Please check and revise the typo and/or reanalyze results accordingly.
4. The discussion point made in the third paragraph of section 4 (lines 97-110) strikes me as a bit of a leap. This is a non-randomized, pilot trial and so it seems early to say that results would indicate the entire ED classification system should be reconsidered!
In some prior mirror exposure studies, a history of self-harm or suicidality was determined to be a contraindication for the interventions. Given the authors findings about baseline depression being a mediator of treatment outcome, the discussion would be strengthened by elaborating on the clinical implications of this and how this differs from or dovetails from prior research in body exposure.
--
In summary, the study addresses a very important topic and has the potential to make an important contribution to the literature. However, several elements of the manuscript are in need of clarification and revision.
Reviewer 2 Report
The manuscript is interesting and it tackle an important topic in the eating disorder’s field.
I have only some question and suggestion to make:
1) It is not reported if the study was approved by an ethics committee and I ask to clear this point. 2) I have some doubts as far as the choice to add OSFED to the others ED diagnostic categories according to their clinical presentation for the statistical analysis. In this way the AN group of 26 patients becomes of 46 subjects with the addition of OSFED patients and this might influence results. Moreover, the Authors declare that they used DSM IV criteria till 2017 and DSM 5 criteria after 2017: in DSM IV there was not the OSFED category but the EDNOS one. Please clear these points. In general, to put together different ED diagnosis as far body image is not recommended because ED patients have different attitude towards their own body image. 3) The AN group’s mean BMI is far away from what is considered underweight and the Authors need to explain better this point as they did in the limits of the paper. The BMI is due to a less severe pathology or to an initial stage of the pathology (early diagnosis?) or to a final stage of treatment? Do people participating to the study was previously treated for their ED? Which are the exclusion criteria for the Positive Body Exposure treatment: the severity of ED symptoms? BMI? The presence of other psychiatric pathology? What else? 4) Even though authors refer to their publications for a more thorough description, because they report a large range of session delivered according to the patient’s needs, it would be necessary at least to explain how many sessions per week were delivered and the entire duration of the protocol (how many weeks or months?). 5) As far as the results, it is quite strange that AN group scored lower than BED at BAT, and lower than BN with regard to the EDE-Q total and eating concern subscale. Please explain your findings that are quite different from literature. 6) The page numbering is correct till page 7 than starts again from page 1: I hope it is only a mistake of numbering. 7) Page 4 – Paragraph 2.3: the first sentence is written twice.Author Response
Please see the attachment
